# Can a Difference in Gestational Age According to Biparietal Diameter and Abdominal Circumference Predict Intrapartum Placental Abruption?

**DOI:** 10.3390/jcm10112413

**Published:** 2021-05-29

**Authors:** Jee-Youn Hong, Jin-Ha Kim, Seo-yeon Kim, Ji-Hee Sung, Suk-Joo Choi, Soo-young Oh, Cheong-Rae Roh

**Affiliations:** 1Department of Obstetrics and Gynecology, Samsung Medical Center, Sungkyunkwan University School of Medicine, Seoul 06351, Korea; lubhide@hanmail.net (J.-Y.H.); jinha1114@hanmail.net (J.-H.K.); obgysung@gmail.com (J.-H.S.); drmaxmix.choi@samsung.com (S.-J.C.); cr.roh@samsung.com (C.-R.R.); 2Department of Obstetrics and Gynecology, Kangbuk Samsung Hospital, Sungkyunkwan University School of Medicine, Seoul 03181, Korea; seoyeon87.kim@samsung.com

**Keywords:** placental abruption, biparietal diameter, abdominal circumference, ultrasound examination

## Abstract

This study aimed to investigate whether a difference in gestational age according to biparietal diameter (BPD) and abdominal circumference (AC) could be a clinically useful predictor of placental abruption during the intrapartum period. This retrospective cohort study was based on singletons who were delivered after 32 + 0 weeks between July 2015 and July 2020. We only included cases with at least two antepartum sonographies available within 4 weeks of delivery (*n* = 2790). We divided the study population into two groups according to the presence or absence of placental abruption and compared the clinical variables. The incidence of placental abruption was 2.0% (56/2790) and was associated with an older maternal age, a higher rate of preeclampsia, and being small for the gestational age. A difference of >2 weeks in gestational age according to BPD and AC occurred at a higher rate in the placental abruption group compared to the no abruption group (>2 weeks, 21.4% (12/56) vs. 7.5% (205/2734), *p* < 0.001; >3 weeks, 12.5% (7/56) vs. 2.0% (56/2734), *p* < 0.001). Logistic regression analysis revealed that the differences of >2 weeks and >3 weeks were both independent risk factors for placental abruption (odds ratio (OR) (95% confidence interval), 2.289 (1.140–4.600) and 3.918 (1.517–9.771), respectively) after adjusting for maternal age, preeclampsia, and small for gestational age births. We identified that a difference in gestational age of >2 weeks between BPD and AC could be an independent predictor of placental abruption.

## 1. Introduction

Placental abruption, defined as premature detachment of the placenta from the uterine wall, occurs in 0.4–1.3% of all pregnancies [1,2]. The etiology of placental abruption is not fully understood [3]. However, it is suspected in women with symptoms such as vaginal bleeding or abdominal pain. The final diagnosis is mainly based on placental inspection after delivery. The perinatal mortality rate varies between 2% and 67%, depending on gestational age, fetal weight, and the degree of abruption [1]. Additionally, placental abruption results in an increased frequency of low birth weight and preterm deliveries [4,5].

As maternal age has become a worldwide trend, the risk of placental abruption and adverse perinatal outcomes is increasing globally, including in South Korea [6,7,8,9]. The risk factors associated with placental abruption are multifactorial and include advanced maternal age, chronic hypertension with or without preeclampsia, premature rupture of membranes (PROM), preterm birth, and fetal growth restriction (FGR) [2,7]. One study of placental abruption in term pregnancies reported that FGR had an odds ratio (OR) of 4.0 and a 95% confidence interval (CI) of 2.3–6.8. This study also found that mild and severe pregnancy-induced hypertension (OR (95% CI), 2.4 (1.5–3.8) in mild; 4.6 (2.2–9.4) in severe) were found to be independently associated with the occurrence of placental abruption [7]. According to a meta-analysis of studies on placental abruption, patients with chronic hypertension were more than three times more likely to develop placental abruption than normotensive patients (OR (95% CI), 3.13 (2.04–4.80)) [10]. It was also suggested that pregnancies with preterm PROM had an increased risk of placental abruption (OR (95% CI), 6.1 (4.1–9.0)) [11].

Studies on FGR as a risk factor for placental abruption have shown that the overall OR for placental abruption in FGR groups compared to control groups were 2.06 (95% CI, 1.57–2.55) [2,12,13,14,15]. A study of 7,508,655 US birth records assembled by the National Center for Health Statistics reported that mothers of infants in the lowest weight centile (<1% adjusted for gestational age) were almost nine times more likely to have placental abruption than those in the heaviest (≥90%) birth weight centile. This relative risk declined progressively with higher birth weight centiles [1]. However, the definitions of FGR in national guidelines are not consistent [16]. In the United States and France, a birthweight or estimated fetal weight (EFW) lower than the 10th population centile is used [17,18], while in the United Kingdom, New Zealand, and Canada, an EFW and abdominal circumference (AC) lower than the 10th population centile are used [16]. Indeed, an AC lower than the 10th population centile has been reported to be a strong predictor of FGR despite the heterogeneity of the study designs [19,20,21]. Given the unpredictability and critical significance of placental abruption, identifying clinically useful predictive factors during the intrapartum period is necessary for detecting at-risk individuals.

Consequently, we hypothesized that fetuses with greater differences in biparietal diameter (BPD) and AC would be at an increased risk of adverse outcomes, including placental abruption. We investigated the association between placental abruption and the difference in gestational age according to BPD and AC in women expecting vaginal delivery and to determine whether this difference could be an independent risk factor for placental abruption using multivariate analysis.

## 2. Materials and Methods

This study was based on consecutive singletons born after 32 + 0 weeks at our institution between July 2015 and July 2020. The retrospective study protocol was approved by the institutional review board of our institution (approval number 2021-01-103). Although 4736 women delivered during the study period, we only included cases with at least two antepartum sonographies available within 4 weeks of delivery, in which BPD, AC, and EFW were recorded. Pregnancies with multiple gestations, major fetal anomalies, any planned cesarean section due to several reasons, including abnormal fetal presentation, previous cesarean section or myomectomy, and placenta previa, were excluded. Finally, 2790 singleton pregnancies met the inclusion criteria for this study (Figure 1).

The clinical variables used in this study were derived from detailed computerized information on maternal and neonatal medical records. All ultrasound examinations were performed using one of the following ultrasound units: WS80A (Samsung Medison Co., Ltd., Seoul, Korea), Voluson E8, or Voluson E10 (GE Healthcare Korea, Seoul, Korea). Hadlock tables were used to determine the corresponding gestational age for BPD, AC, and EFW [22]. Small for gestational age (SGA) was defined as at least two antepartum EFWs (performed within 4 weeks of delivery) calculated below the 10th, 5th, and 3rd centile based on nomograms published by national data from the Korean Health Insurance Review and Assessment Service (2009) and the NICHD fetal growth studies [23]. The mean difference in gestational age for BPD and AC was calculated based on two sonographic examinations performed within 4 weeks of delivery. The diagnosis of placental abruption was based on gross clinical examination of the placenta by the attending physician at the time of delivery, which was documented in the medical record. Women were regarded as placental abruption cases if they satisfied at least one of following four clinical criteria: (i) patients presenting with vaginal bleeding or abdominal pain or both; (ii) non-reassuring fetal status with bradycardia was documented; (iii) placental abruption diagnosed on prenatal ultrasound; and (iv) delivered placenta showing evidence of clinically significant retro-placental bleeding or clots. We divided the study into two groups according to the presence or absence of placental abruption and compared the following variables: baseline characteristics, which included maternal age, body mass index (BMI) at delivery, past obstetric history (gravidity and parity, history of prior pregnancy complicated by stillbirth or preterm delivery, and history of conception by assisted reproductive technology), past medical history (history of pre-gestational diabetes mellitus, pre-pregnancy hypertension, preeclampsia, and cigarette smoking), and current pregnancy complication (gestational diabetes mellitus and preeclampsia). The sonographic features included oligohydramnios, EFW, and any difference in gestational age according to BPD and AC (>2 weeks or >3 weeks).

For delivery outcomes, the mode of delivery and indication for cesarean delivery were obtained. Indications for cesarean delivery were classified as induction failure, failure to progress, fetal distress, maternal complication (uncontrolled blood pressure or symptoms due to pregnancy-induced hypertension), or maternal request. The diagnosis of morphologically abnormal placenta (circumvallate, succenturiate, globular, or placentomegaly) and umbilical cords (marginal, velamentous, true knot, hypocolied, hypercoiled, thin, or long) was based on clinical data and gross pathology. The major neonatal outcomes included birth weight, 1-min and 5-min Apgar scores, cord blood analysis results, neonatal death, and neonatal intensive care unit (NICU) admission. Neonatal death was defined as death within 28 days of birth.

The Student’s t-test was used for continuous variables, and the chi square or Fisher’s exact test was used for categorical variables. Multivariate logistic regression was performed to determine whether a difference in gestational age according to BPD and AC of >2 weeks could be an independent risk factor for placental abruption. A receiver operating characteristic (ROC) curve was constructed to determine the area under the curve (AUC) for a difference in gestational age according to BPD and AC as a predictor of placental abruption. A value of *p* < 0.05 was considered significant. SPSS Statistics 27 (SPSS Inc., Chicago, IL, USA) was used for statistical analysis.

## 3. Results

In this study population, the incidence of abruption was 2.0% (56/2790). Among the placental abruption cases, 48.2% (27/56) were suspected clinically before birth based on symptoms (vaginal bleeding or abdominal pain or both, 3.3% (9/27)), signs (fetal distress, 70.3% (19/27)), and ultrasound examinations (18.5%, (5/27)). There were no statistically significant associations between the antenatal symptoms or signs and the difference in gestational age according to BPD and AC. However, 51.7% (29/56) were asymptomatic and only confirmed to be placental abruption after birth.

Table 1 summarizes the baseline characteristics, sonographic features, and delivery outcomes according to the presence or absence of placental abruption. The mean maternal age was higher in the placental abruption group than in the no abruption group (34.6 ± 3.6 years vs. 33.3 ± 3.7 years, *p* = 0.001), and the incidence of preeclampsia was higher (12.5% vs. 4.1%, *p* = 0.002). Other baseline characteristics, including nulliparity and BMI, were not different between the two groups. The rate of SGA (below the 10th, 5th, and 3rd centile) was higher in the placental abruption group than in the no abruption group (32.1% vs. 7.8% for the 10th centile; 25.0% vs. 4.1% for the 5th centile; and 23.2% vs. 3.1% for the 3rd centile), whereas the rate of appropriate for gestational age (AGA) births was lower (62.5% vs. 84.9%, *p* < 0.001). The incidence of a difference between gestational age according to BPD and AC (>2 weeks and >3 weeks) was significantly higher in the placental abruption group compared to the no abruption group (21.4% vs. 7.5% for >2 weeks; and 12.5% vs. 2.0% for >3 weeks). As expected, the rates of preterm delivery and cesarean section due to fetal distress were higher in the abruption group (35.7% vs. 5.7%, *p* < 0.001; and 30.4% vs. 2.6%, *p* < 0.001, respectively). Placental and umbilical cord abnormalities were higher in the placental abruption group (5.4% vs. 1.9%, *p* = 0.093), although the difference was not statistically significant.

Table 2 shows the neonatal outcomes according to the presence or absence of placental abruption. As expected, low Apgar scores and low birth weights were prevalent in the placental abruption group. Moreover, the rate of umbilical cord pH lower than 7.1 (17.9% vs. 1.3%, *p* < 0.001), NICU admission (48.2% vs. 11.9%, *p* < 0.001), and neonatal death (3.6% vs. 0.1%, *p* = 0.004) were also higher in the placental abruption group. Among the neonates who were admitted to NICU, 51.8% (14/27) had symptoms of placental abruption; these were vaginal bleeding or abdominal pain (14.3%, 2/14), fetal bradycardia (78.6%, 11/14), and sonographic diagnosis (28.6%, 4/14).

Table 3 shows the logistic regression analysis indicating that a difference in gestational age according to BPD and AC was an independent risk factor for placental abruption (OR (95% CI) > 2 weeks, 2.289 (1.140–4.600); >3 weeks, 3.918 (1.517–9.771)) after adjusting for maternal age, preeclampsia, and SGA (<10th centile). We also performed the same regression analysis using different cut-offs for SGA, below the 5th centile or 3rd centile. As expected, the severity of SGA was significantly associated with higher risk for placental abruption. Of note, a difference in gestational age according to BPD and AC (>2 weeks or >3 weeks) remained as an independent risk factor for placental abruption after adjusting for a severe degree of SGA (<5th or <3rd centile).

Table 4 shows the diagnostic performance analysis. The positive predictive value (PPV) and negative predictive value (NPV) for placental abruption was 5.5% and 98.3% for a difference of >2 weeks and 11.1% and 98.2% for a difference of >3 weeks, respectively.

Overall, the ROC curve analysis showed that the AUC for predicting placental abruption using the difference in gestational age according to BPD and AC was 0.638 (*p* < 0.001, Figure 2A). In addition, the AUC for predicting placental abruption using the EFW performed at the last scan was 0.702 (*p* < 0.001, Figure 2B).

## 4. Discussion

In the present study, we demonstrated that a difference of >2 weeks in gestational age according to BPD and AC could be an independent predictor of placental abruption after adjusting for confounders, including maternal age, SGA (<10th, 5th, and 3rd centile), and preeclampsia. Specifically, a difference of >2 weeks or >3 weeks in gestational age between BPD and AC was associated with greater odds of placental abruption by approximately 2.3 or 3.9, respectively. These results suggest that a greater difference in gestational age between BPD and AC was associated with a higher risk of placental abruption. In fact, in our analysis, the increased risk ratio for placental abruption associated with a difference in gestational age according to BPD and AC was comparable with that of well-known risk factors such as preeclampsia (OR range, 2.18–2.25) or SGA (OR range, 4.81–4.66). Our data also showed that the overall AUC for predicting placental abruption by mean difference in gestational age according to BPD and AC was 0.638, which is relatively low. However, considering the low prevalence but critical and unpredictable features of placental abruption, a PPV ranging from 5.5% to 11.1% for a difference of >2 weeks could be considered meaningful and clinically valuable.

The most commonly adopted definition of FGR is an EFW below the 10th centile [16]. However, a small AC below the 10th centile or 5th centile is also used as diagnostic criteria and is a sensitive predictor of FGR [19,20,21]. According to the Delphi procedure, a consensus definition of fetal growth restriction, two solitary parameters (AC or EFW < 3rd centile), and four contributory parameters (EFW or AC <  10th centile, AC or EFW crossing centiles by  >  two quartiles on growth charts, and cerebro-placental ratio  <5th centile or UA-PI > 95th centile) were defined for late FGR (≥32 weeks) [24]. In a retrospective study of the ultrasounds of 1594 pregnant women at ≥36 weeks by Rad et al., an AC lower than the 10th centile a had higher sensitivity for predicting SGA in neonates (sensitivity, 64.0% for AC < 10th centile vs. 50.6% for EFW < 10th centile), with a similar PPV (81.3% for AC < 10th centile vs. 83.8% for EFW < 10th centile) compared with an EFW lower than the 10th centile [19]. Additionally, using AC lower than the 10th centile and EFW lower than the 10th centile together has a higher sensitivity and comparable specificity to a screening tool for FGR with a low false-positive rate (sensitivity 67.5%, false-positive rate 3.3%, PPV 80.3, and AUC 0.821) [19].

Indeed, several studies have indicated that a small AC and low EFW are important factors for predicting adverse perinatal outcomes. According to the Prospective Observational Trial to Optimize Health in Intrauterine Growth Restriction (PORTO) study conducted by the Perinatal Ireland Research Consortium, which included 1116 preterm fetuses with an EFW lower than the 10th centile, an EFW lower than the 3rd centile alone, and an EFW or AC lower than the 10th centile, 5th centile, or 3rd centile, combined with abnormal umbilical artery Doppler velocimetry, were significantly correlated with the NICU admission rate and adverse perinatal outcomes (defined as a composite outcome of intraventricular hemorrhage, periventricular leukomalacia, hypoxic ischemic encephalopathy, necrotizing enterocolitis, bronchopulmonary dysplasia, sepsis, and death) [25]. In a retrospective study including 2237 term and 455 preterm patients, an AC lower than the 5th centile was also found to be associated with composite morbidity and mortality defined as hypoxic-ischemia encephalopathy, periventricular leukomalacia, necrotizing enterocolitis, sepsis, renal failure, or death (adjusted OR (95% CI), 3.77 (1.35–10.5) for term deliveries; and 3.46 (1.89–6.32) for preterm deliveries) [26]. According to a retrospective study including 592 pregnant women from 28 + 0 to 33 + 6 weeks gestation, small AC was associated with a higher rate of indicated preterm delivery compared to the normal AC group (RR (95% CI), 3.7 (1.8–75), *p* = 0.002) [27]. In our study, we included 2790 women >32 + 0 weeks gestation who were expecting to deliver vaginally and demonstrated a significant association between a smaller gestational age according to AC (compared to BPD) and placental abruption, which is one of the important causes of adverse neonatal outcomes and an indication of preterm delivery. Our findings of a four-fold increased risk of placental abruption in pregnancies with SGA fetuses is in a similar range as Kramer et al. and Lindqvist et al. [12,28]. According to a study including 223,341 singleton deliveries, the rate of SGA was significantly higher in pregnancies complicated with placental abruption (15.6% vs. 11.0, *p* <0.001) [5]. A recent systemic review indicated that IUGR/SGA had an elevated risk of abruption, ranging between 1.3 and 17.4 [29].

Traditionally, a smaller AC relative to fetal head size was considered an index of asymmetric FGR, reflecting brain sparing [30,31]. Moreover, asymmetric FGR is associated with placental insufficiency [32]. According to a study of pathologic placental examinations in 156 late-onset SGA births, placental insufficiency due to maternal under-perfusion was observed in two-thirds of the term SGA neonates despite normal umbilical artery Doppler ultrasounds and was associated with adverse perinatal outcomes (low birth weight, birth weight percentile, and duration of NICU stay) [33]. The sonographic assessment of brain sparing in relation to adverse pregnancy outcomes has been studied to some extent. A systemic review and meta-analysis showed that a middle cerebral artery Doppler in FGR was associated with composite adverse perinatal outcomes, including a low Apgar score, NICU admission, perinatal morbidity, and mortality (positive likelihood ratio, 9.32 (3.91–22.19); negative likelihood ratio, 0.53 (0.43–0.65)) [34]. The transverse cerebellar diameter (TCD)-to-AC ratio previously used for fetal growth assessment was also utilized as a parameter of brain sparing (asymmetric FGR) [35,36]. According to this study, which included 473 SGA births who underwent ultrasounds at 24 + 0 through 28 + 0 weeks, the TCD-to-AC ratio was associated with a higher risk of developing maternal placenta syndrome (hypertensive disorders of pregnancy and placental abruption) [37]. In the present study, we used the difference in gestational age according to BPD and AC as a marker of brain sparing and examined this parameter as a predictor of placental abruption since these biometric measurements are routinely performed with every fetal ultrasound.

During normal pregnancy, the physiological remodeling of the spiral arteries results in the perfusion of the intervillous space with a dramatic increase in terminal luminal diameter [38]. However, in pregnancy disorders associated with placental ischemia, changes in the spiral arteries and uterine vessels result in reduced delivery of nutrients and oxygen to the intervillous space [39]. These similar abnormal modifications are found in pregnancies with preeclampsia, FGR [40], in one-third of preterm births [41], and in some normal pregnancies [42]. Others have suggested that placental pathophysiology in FGR is an intermediated form between normal and preeclamptic pregnancies [43]. Placental abruption, as well as preeclampsia and FGR, may be considered placental ischemia-related disorders. Moreover, FGR and preeclampsia have been associated with placental abruption. Consistent with previous studies, we confirmed that SGA and preeclampsia were associated with a higher risk of placental abruption. AC is considered the primary parameter associated with FGR [44,45,46,47]. According to a systemic review and meta-analysis of third-trimester ultrasounds, a small AC (<10th centile) and EFW (<10th centile) had similar diagnostic capacity to predict FGR [46]. Another study also proposed that AC is strongly associated with fetal nutritional status due to the reflection of liver size and abdominal subcutaneous fat storage [47]. Recently, Sheth and Glantz suggested that a small AC may be a result of the initial depletion of fetal glycogen stores, whereas a small femur length may be consistent biometric parameter that indicates long-stand deprivation [26]. On the other hand, small AC in terms of asymmetric FGR may reflect the brain-sparing effect caused by placental insufficiency and altered adaptation of fetal cerebral hemodynamics [30,45]. In our study, we found that a greater difference between BPD and AC could reflect brain sparing and was clinically associated with a higher risk of placental abruption.

Our study has several limitations. First, this was a retrospective cohort study based on patients in a tertiary hospital with a relatively high prevalence of high-risk pregnancies, limiting generalizability to low-risk pregnant women. In fact, the prevalence of placental abruption in our study population was 2%, which was somewhat higher than that in the general population [1,2]. Secondly, due to the long study period, several different physicians measured the BPD, AC, and EFW. Thus, to overcome this limitation, we used the mean difference in gestational age for BPD and AC based on two sonographic examinations, which were available within 4 weeks of delivery. This strict inclusion criteria inevitably made the number of target groups about three-quarters of the total and this is likely to act as another selection bias in reflecting the high-risk pregnancies.

In conclusion, after 32 weeks gestational age, a >2-week difference in gestational age according to BPD and AC can be used to predict intrapartum placental abruption. Considering that only 48.2% had antenatal symptoms of placental abruption, our study suggests that once there is clinical suspicion for or diagnosis of a small AC on sonography, clinicians should carefully evaluate the fetal monitor during the intrapartum period for indications of placental abruption regardless of whether FGR is detected. We believe our findings may be clinically useful in hospitals or countries where small AC alone is not incorporated into the diagnostic criteria for FGR.

## Figures and Tables

**Figure 1 jcm-10-02413-f001:**
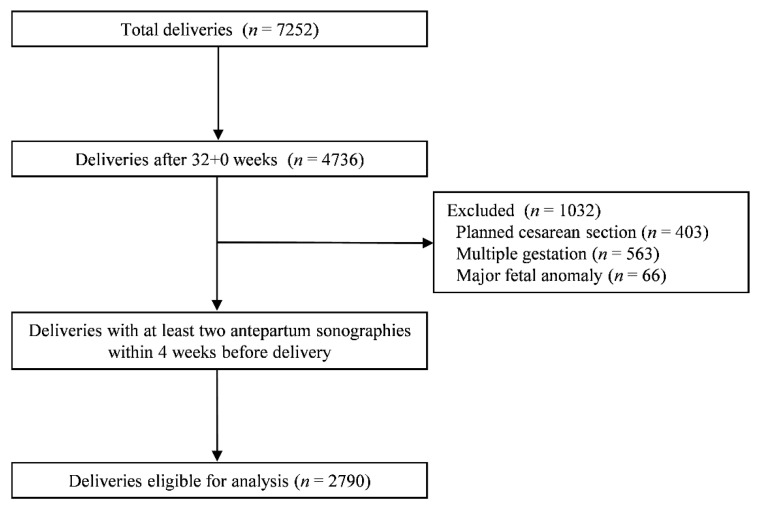
Flowchart of the study population.

**Figure 2 jcm-10-02413-f002:**
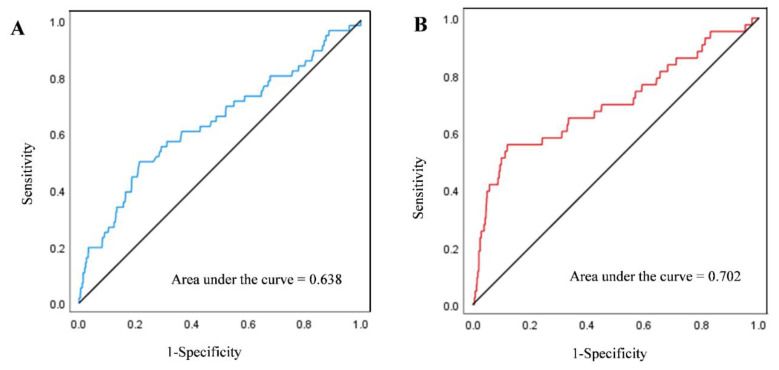
(**A**) Receiver operating characteristic curve of the association between mean difference in gestational age between BPD and AC and placental abruption. (**B**) Receiver operating characteristic curve of the association between EFW at last scan and placental abruption. Abbreviations: BPD, biparietal diameter; AC, abdominal circumference; EFW, estimated fetal weight.

**Table 1 jcm-10-02413-t001:** Comparison of the baseline characteristics, sonographic features, and delivery outcomes according to placental abruption.

	Abruption (+) (*n* = 56)	Abruption (−) (*n* = 2734)	*p*-Value
**Baseline characteristics**			
Maternal age, years (mean ± SD)	34.6 ± 3.6	33.3 ± 3.7	0.007
Nulliparity, *n* (%)	31 (55.4)	1764 (64.5)	0.156
Smoker, *n* (%)	0 (0.0)	22 (0.8)	1.000
BMI (kg/m^2^) at delivery	26.4 ± 3.8	26.2 ± 3.5	0.826
Assisted reproduction techniques, *n* (%)	3 (5.4)	154 (5.6)	1.000
**Past obstetric history**			
Previous stillbirth at >20 weeks, *n* (%)	2 (3.6)	24 (0.9)	0.095
Previous preterm delivery, *n* (%)	3 (5.4)	78 (2.9)	0.221
**Past medical history**			
Previous HTN, preeclampsia, *n* (%)	2 (3.6)	28 (1.0)	0.121
Previous DM, *n* (%)	1 (1.8)	29 (1.1)	0.457
**Current pregnancy complications**			
Gestational diabetes mellitus, *n* (%)	6 (10.7)	245 (9.0)	0.65
Preeclampsia, *n* (%)	7 (12.5)	112 (4.1)	0.002
**Sonographic features**			
Oligohydramnios, *n* (%)	0 (0.0)	50 (1.9)	0.625
EFW			
SGA (<10th centile), *n* (%)	18 (32.1)	212 (7.8)	<0.001
SGA (<5th centile), *n* (%)	14 (25.0)	112 (4.1)	<0.001
SGA (<3rd centile), *n* (%)	13 (23.2)	86 (3.1)	<0.001
AGA, *n* (%)	35 (62.5)	2320 (84.9)	<0.001
LGA, *n* (%)	2 (3.6)	34 (1.2)	0.162
Difference in BPD and AC gestational age, *n* (%)			
>2 weeks	12 (21.4)	205 (7.5)	<0.001
>3 weeks	7 (12.5)	56 (2.0)	<0.001
**Delivery outcomes**			
Gestational age at delivery, weeks	37.4 ± 2.4	39.3 ± 1.4	<0.001
Preterm, 32–36 weeks, *n* (%)	20 (35.7)	156 (5.7)	<0.001
Induction or augmentation, *n* (%)	34 (60.7)	1464 (53.5)	0.287
Mode of delivery			<0.001
Normal vaginal devliery, *n* (%)	33 (58.9)	2155 (78.8)	
Cesarean delivery, *n* (%)	23 (41.1)	579 (21.2)	
Induction failure	1 (1.8)	187 (6.8)	0.178
Fetal distress	17 (30.4)	70 (2.6)	<0.001
Failure to progress	1 (1.8)	209 (7.6)	0.123
Other	4 (7.1)	113 (4.1)	0.295
Placental and umbilical cord abnormality, *n* (%)	3 (5.4)	51 (1.9)	0.093

Values are presented as the mean ± standard deviation or *n* (%). The difference in BPD and AC gestational age was calculated using the Hadlock table as follows: gestational age in weeks according to BPD (gestational weeks + days/7)—gestational age in weeks according to abdominal circumference (gestational weeks + days/7). BMI, body mass index; EFW, estimated fetal weight (by Hadlock formula); SGA, small for gestational age; AGA, appropriate for gestational age; LGA, large for gestational age (>90th centile); HTN, hypertension; DM, diabetes mellitus; BPD, biparietal diameter; AC, abdominal circumference.

**Table 2 jcm-10-02413-t002:** Neonatal outcomes according to placental abruption.

	Abruption (+) (*n* = 56)	Abruption (−) (*n* = 2734)	*p*-Value
Birth weight, kg (mean ± SD)	2.7 ± 0.7	3.2 ± 0.5	<0.001
Low Apgar score, 1 min (<4)	9 (16.1)	16 (0.6)	<0.001
Low Apgar score, 5 min (<7)	4 (7.1)	10 (0.4)	<0.001
Cord pH < 7.1	5 (17.9)	25 (1.3)	<0.001
Base excess of cord blood (mmol/ℓ)	−6.6 ± 5.4	−4.4 ± 2.7	0.042
Base excess of cord blood <−12 (mmol/ℓ)	4 (14.3)	22 (1.2)	<0.001
NICU admission, *n* (%)	27 (48.2)	324 (11.9)	<0.001
Duration of NICU stay, days (mean ± SD)	12.9 ± 44.0	1.3 ± 9.6	0.054
Neonatal death, *n* (%)	2 (3.6)	3 (0.1)	0.004

Data are presented as the mean ± standard deviation or *n* (%). NICU, neonatal intensive care unit.

**Table 3 jcm-10-02413-t003:** Logistic regression analysis of the risk factors for placental abruption.

	Adjusted Odds Ratio	95% CI	*p*-Value		Adjusted Odds Ratio	95% CI	*p*-Value
Maternal age	1.116	1.038–1.199	0.003	Maternal age	1.114	1.036–1.197	0.004
SGA (<10th centile)	4.819	2.606–8.909	<0.001	SGA (<10th centile)	4.669	2.505–8.702	<0.001
Preeclampsia	2.183	0.936–5.088	0.071	Preeclampsia	2.258	0.962–5.302	0.061
Difference of >2 weeks between BPD and AC gestational age	2.289	1.140–4.600	0.020	Difference of >3 weeks between BPD and AC gestational age	3.918	1.517–9.771	0.003
Maternal age	1.121	1.042–1.206	0.002	Maternal age	1.117	1.039–1.202	0.003
SGA (<5th centile)	6.516	3.228–13.154	<0.001	SGA (<5th centile)	6.192	3.012–12.729	<0.001
Preeclampsia	2.098	0.891–4.942	0.071	Preeclampsia	2.178	0.919–5.162	0.077
Difference of >2 weeks between BPD and AC gestational age	2.067	1.003–4.258	0.049	Difference of >3 weeks between BPD and AC gestational age	3.186	1.218–8.336	0.018
Maternal age	1.118	1.039–1.204	0.003	Maternal age	1.117	1.037–1.202	0.003
SGA (<3rd centile)	7.513	3.616–15.608	<0.001	SGA (<3rd centile)	7.164	3.395–15.119	<0.001
Preeclampsia	2.068	0.871–4.906	0.099	Preeclampsia	2.129	0.891–5.089	0.089
Difference of >2 weeks between BPD and AC gestational age	2.039	0.983–4.227	0.056	Difference of >3 weeks between BPD and AC gestational age	3.209	1.218–8.453	0.018

Adjusted odds ratio (statistically significant after adjusting for multiple comparisons). The difference in BPD and AC gestational age was calculated using the Hadlock table as follows: gestational age in weeks according to BPD (gestational weeks + days/7)—gestational age in weeks according to abdominal circumference (gestational weeks + days/7). SGA, small for gestational age; CI, confidence interval; BPD, biparietal diameter; AC, abdominal circumference.

**Table 4 jcm-10-02413-t004:** Diagnostic performance for prediction of placental abruption.

	Difference of >2 Weeks between BPD and AC Gestational Age	Difference of >3 Weeks between BPD and AC Gestational Age
Positive predictive value	5.5%	11.1%
Negative predictive value	98.3%	98.2%
Sensitivity	21.4%	12.5%
Specificity	92.5%	98.0%

BPD, biparietal diameter; AC, abdominal circumference.

## Data Availability

The datasets analyzed during the current study are available from the corresponding author on reasonable request.

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
