# Peer review of "Can a Difference in Gestational Age According to Biparietal Diameter and Abdominal Circumference Predict Intrapartum Placental Abruption?"

_jcm, 2021, doi:10.3390/jcm10112413_

Round 1

Reviewer 1 Report

The definition of the outcome ‘placental abruption’ is unclear: Line 107: ‘Based on gross clinical examination of the placenta by the attending physician at the time of the delivery’. Minimum size of blodclot?/depression on the placenta?/haemorhage?/interobserver differences? This is important as more than 50% of the women had no symptoms of abruption. Is it possible, that the findings have developed after delivery of the child - before delivery of the placenta?

Only women with ‘at least two antepartum sonographies available within 4 weeks before delivery’ were included (line 89). Error in Figure 1: 2790 women with two ultrasound scans (not 3704 women as noted) Thereby only about 75% of the population were included: Why were these women scanned compared to the non-scanned population? Highrisk pregnancies with hypertension, preeclampsia, clinical SGA? Was Doppler scan for brain sparing never included?

Differences in gestational age for BPD and AC were calculated based on the mean of two sonographic examinations performed within 4 weeks before delivery (line line 105), and SGA were based on two antepartum EFW less than 10th percentile. Why were the women scanned twice within 4 weeks? Who performed the scans (trained sonographers with two or more weeks in between, or untrained and trained sonographers within days)?

The difference in gestational age according to BPD and AC are used (> 2 weeks and > 3 weeks). Did you consider using AC percentile deviation instead/too (as discussed in the discussion section). Calculations of the diagnostic performance for prediction of placental abruption are less relevant – as >50% didn’t have any symptoms. Further, the sensitivity and the PPV is low, even in a high-risk population (2% abruption). ROC curve (Figure 2) is of less interest, as AUC is so low

Table 3: The adjusted OR for SGA are 4,7-4,8 and in the discussion different screening tools for FGR are discussed (EFW, AC). Consider showing your data on the degree of SGA/FGR, as the EFW has already been calculated (eg. <2½th , <5th, <10th percentiles) instead of only SGA (<10th percentile). Is the degree of FGR a predictor for abruption ?

Only 48,2% (27 women) of women with abruption had symptoms or signs – 29 women were asymptomatic (line 177-78) and probably without clinical relevance. It would be of more clinical relevance if data regarding the symptomatic women were shown separately in the Result section – including the association between the difference in gestational age according to BPD and AC. Eg. 27 children were admitted to NICU – the 27 children where the mothers had symptoms of abruption?

Line 91-92. Every woman with planned CS were excluded – or only planned CS due to the mentioned indications?

Line 110-116/Table 1: Prior history of placental abruption is not included?

Lines 97-135 and lines 136-174 are a repetition.

Table 2: Only 9 children showed sign of asphyxia (either low apgar score after 5 min or cord pH<7,10), however, 27 children went to the NICU admission. Was It because of prematurity? Or neonatal anaemia/do you have information about hemoglobin level?

Line 335-37: could reflect brain sparing instead of reflected higher brainsparing

Author Response

Please see the attachment below.

Reviewer 2 Report

It has already been reported that FGR is a risk factor for Pacental abruption.
A difference in gestational age according to biparietal diameter and abdominal circumference is synonymous with FGR.
It's just a paraphrase of FGR. Therefore,
I cannot accept this study design.

Author Response

Dear Professor, 

We would like to thank you for giving a chance to revise our manuscript entitled “Can a difference in gestational age according to biparietal diameter and abdominal circumference predict intrapartum placental abruption?”

We appreciate the time and effort that you and the reviewers have dedicated to providing your valuable feedback on our manuscript. We are grateful to the reviewers for their insightful comments. We have been able to incorporate changes to reflect most of the suggestions provided by the reviewers. We have highlighted the changes in yellow color within the manuscript.

We hope that our corrections are a suitable response to the reviewers’ comments.

Again, thank you very much for your consideration and hope for a favorable response.

Sincerely yours,

Reviewer 3 Report

The authors propose the use of discrepancy in fetal growth according to abdominal circumference (AC) and biparietal diameter (BPD) (GA_BPD-GA_AC) as for placental abruption. The outcome studied is important, the manuscript is in general well presented. There are methodological aspects which are problematic to conclude that GA_BPD-GA_AC adds something to previously known risk factors.

Major:

  • The significance of GA_BPD-GA_AC as in independent risk factor for placental abruption in addition to the standard definition of SGA was not properly assessed. SGA was defined as “at least two antepartum EFWs below the 10th centile” (not clear if 2 out of any during pregnancy or the same last two used for GA_BPD-GA_AC). This is key because more recent ultrasound scans are more predictive of adverse outcomes. Then, while the SGA definition uses “any” to combine information from multiple scans, GA_BPD-GA_AC is based on averaging of the data over the last two scans. Therefore, to see if indeed the discrepancy between growth by AC and BPD brings any new insight relative to EFW the two variables need to be compared in the same way and at the same gestational age. I would like to see an ROC curve of GA_BPD-GA_AC at last scan vs ROC of the EFW at last scan, and the same for the average of those predictors over the last two scans. If there are no differences in ROC, there is no new insight that was obtained. If the authors want to binaries the predictors, that is fine, but use an EFW cut-off that will give the same false positive rate for both variables (again, based on the same scans).

  • Lines 98-106 seem to be repeated in 137-145

 Minor

  • The authors state in the results: “The mean maternal age was higher in the placental abruption group than in the no abruption group (33.3 ± 7 years vs. 34.6 ± 3.6 years, p = 0.001) “. The mean ages should be reported in the same order as the outcome groups (ie. Abruption first).   

Author Response

Please see the attachment below.

Reviewer 4 Report

The authors studied whether a difference in GA according to BPD and AC could be a useful predictor of placental abruption. Moreover, they analyzed in a large group of patients, several variables and identifiied independent risk factor for placental abruption. The study is interesting and deserves publication. However, there are many issues that must be dealt prior to acceptance. Especially, the English language and style must be reviewed in depth.

Introduction:

The authors have gathered complete background literature for this article. Nevertheless, the writing style should be reviewed in order to convey the information more clearly. For example: 

  • Lines 49-50: The passage is not clear
  • Lines 60-62: I am not sure to understand what you meant here 

Materials and methods: 

  • One of the main strengths of this study is the large number of patients enrolled. In addition, you have used a thorough method to collect data about delivery outcomes. Well done.
  • The diagnosis of placental abruption is only based on clinical examination. This is a weakness of your study. Is it possible to obtain the histological diagnosis too? And please, describe the diagnostic criteria of clinical examination more in detail.  
  • ISUOG guidelines recommend the Delphi consensus criteria (Gordjin et al) for the definition of FGR. Did you use these criteria to search for FGR in the patients enrolled? I think that the difference in gestational age between BPD and AC is a secondary way of identifying FGR. You should explain why you made this choice.
  • From line 136 to line 174 you have repeated something you wrote before, so it may be a typing mistake.
  • The statistic analysis is appropriate for the aims of your study.

Results:

  • The results are shown in a clear and correct way.

Discussion:

The summary described in the first part is very clear. However, you should rewrite the subsequent section (from line 262) that is hard to follow and rather unfocused. In my opinion, you should reconsider it starting from these points:

  • You confirmed the importance of FGR diagnosis but you never mentioned Delphi consensus criteria.
  • You largely explained the correlation between FGR and adverse perinatal outcomes. You should focus on placental abruption instead
  • You should compare your results with modern leterature about correaltion between small fetuses (identified with ultrasound examination) and abruptio placentae.
  • You used the difference in GA between BPD and AC as a marker of brain sparing. Please explain why you didn't use Doppler evaluation and support the reliability of your parameter
  • You are suggesting using the difference in GA between BPD and AC to predict abruptio placentae. However, it has a poor predictive capacity. Explain which are the advantages to use it instead of the standard diagnosis of FGR. I know there are different definitions of FGR, but their predictive capacity for abruptio placentae can be assessed. 

In general lines, the paper might be suitable for publication. However, it needs to be reviewed to clarify some conceptual points and adjust the style flaws (I only detailed some of them). 
I, therefore, recommend a major revision.

Author Response

Dear Professor, 

We would like to thank you for giving a chance to revise our manuscript entitled “Can a difference in gestational age according to biparietal diameter and abdominal circumference predict intrapartum placental abruption?”

We appreciate the time and effort that you and the reviewers have dedicated to providing your valuable feedback on our manuscript. We are grateful to the reviewers for their insightful comments. We have been able to incorporate changes to reflect most of the suggestions provided by the reviewers. We have highlighted the changes in yellow color within the manuscript.

We hope that our corrections are a suitable response to the reviewers’ comments.

Again, thank you very much for your consideration and hope for a favorable response.

Sincerely yours,

Please see the attachment below. 

Round 2

Reviewer 2 Report

no comment

Reviewer 4 Report

no comment